# Ultralong 100 ns spin relaxation time in graphite at room temperature

B. G. Márkus [1,2,3], M. Gmitra [4,5], B. Dóra [6], G. Csősz[3], T. Fehér[3], P. Szirmai[7], B. Náfrádi [7], V. Zólyomi[8], L. Forró[1,7], J. Fabian [9] ✉ & F. Simon [2,10] ✉

Graphite has been intensively studied, yet its electron spins dynamics remains an unresolved problem even 70 years after the first experiments. The central quantities, the longitudinal ($T_1$) and transverse ($T_2$) relaxation times were postulated to be equal, mirroring standard metals, but $T_1$ has never been measured for graphite. Here, based on a detailed band structure calculation including spin-orbit coupling, we predict an unexpected behavior of the relaxation times. We find, based on saturation ESR measurements, that $T_1$ is markedly different from $T_2$. Spins injected with perpendicular polarization with respect to the graphene plane have an extraordinarily long lifetime of 100 ns at room temperature. This is ten times more than in the best graphene samples. The spin diffusion length across graphite planes is thus expected to be ultralong, on the scale of ~ 70 $\mu$m, suggesting that thin films of graphite − or multilayer AB graphene stacks − can be excellent platforms for spintronics applications compatible with 2D van der Waals technologies. Finally, we provide a qualitative account of the observed spin relaxation based on the anisotropic spin admixture of the Bloch states in graphite obtained from density functional theory calculations.

Spintronic devices require materials with a suitably long spin-relaxation time, $\tau_s$. Carbon nanomaterials, such as graphite inter-calated compounds[1], graphene[2], fullerenes[3], and carbon nanotubes[4], have been considered[5–7] for spintronics[8–10], as small spin–orbit coupling (SOC) systems with low concentration of magnetic $^{13}$C nuclei which contribute to a long $\tau_s$. However, experimental data and the theory of spin-relaxation in carbon-based materials face critical open questions. Chiefly, the absolute value of $\tau_s$ in graphene is debated with values ranging from 100 ps to 12 ns[11–16], and theoretical investigations suggest an extrinsic origin of the measured short $\tau_s$ values[17].

Contemporary studies, in order to introduce functionality into spintronic devices[18–26], focus on tailoring the SOC in two-dimensional heterostructures with the help of proximity effect[27–30]. Theory predicted a giant spin-relaxation anisotropy in graphene when in contact with a large-SOC material[31] that was subsequently observed in mono- and bilayer graphene[32–37]. This is in contrast with graphene on a SOC-free substrate having a nearly isotropic spin-relaxation[38–41]. It would be even better to have materials with an intrinsic spin-relaxation time anisotropy, which would enable efficient control over the spin trans-port and thereby boost the development of spintronic devices.

[1]Stavropoulos Center for Complex Quantum Matter, Department of Physics and Astronomy, University of Notre Dame, Notre Dame, IN 46556, USA. [2]Institute for Solid State Physics and Optics, Wigner Research Centre for Physics, Budapest H-1525, Hungary. [3]Department of Physics, Institute of Physics, Budapest University of Technology and Economics, Műegyetem rkp. 3., H-1111 Budapest, Hungary. [4]Institute of Physics, Pavol Jozef Šafárik University in Košice, Park Angelinum 9, 040 01 Košice, Slovakia. [5]Institute of Experimental Physics, Slovak Academy of Sciences, Watsonova 47, 04001 Košice, Slovakia. [6]Department of Theoretical Physics, Institute of Physics and MTA-BME Lendület Topology and Correlation Research Group Budapest University of Technology and Economics, Műegyetem rkp. 3., H-1111 Budapest, Hungary. [7]Laboratory of Physics of Complex Matter, École Polytechnique Fédérale de Lausanne, Lausanne CH-1015, Switzerland. [8]STFC Hartree Centre, Daresbury Laboratory, Daresbury Warrington WA4 4AD, UK. [9]Department of Physics, University of Regensburg, 93040 Regensburg, Germany. [10]Department of Physics, Institute of Physics and ELKH-BME Condensed Matter Research Group Budapest University of Technology and Economics, Műegyetem rkp. 3., H-1111 Budapest, Hungary. ✉e-mail: jaroslav.fabian@ur.de; simon.ferenc@ttk.bme.hu

A remarkably simple example of an anisotropic carbon-based material is graphite, which, while being one of the most extensively studied crystalline materials, still holds several puzzles. Specifically, the spin-relaxation, its anisotropy, and the $g$-factor are not yet understood in graphite, and this represents a 70-year-old challenge. As early as 1953, the first spin spectroscopic study of graphite[42, 43] used conduction electron spin resonance (CESR). The CESR linewidth, $\Delta B$, yields directly the spin-decoherence time: $T_2 = (\gamma \Delta B)^{-1}$, where $\gamma/2\pi \approx 28$ GHz/T is the electron gyromagnetic ratio (which is related to the $g$-factor as $|\gamma| = g\mu_B/\hbar$). Magnetic resonance is characterized by two distinct relaxation times, $T_1$ and $T_2$, which denote the relaxation of the components parallel and perpendicular to the external magnetic field, respectively[44]. In zero magnetic field, $T_1 = T_2 = \tau_s$ holds and the latter parameter is measured in spin-injected transport studies.

The ESR linewidth and $g$-factor have a peculiar anisotropy in graphite. $\Delta B$ is about twice as large for a magnetic field perpendicular to the graphene layers (denoted as $\Delta B_\perp$) as compared to when the magnetic field is in the $(a, b)$ plane (denoted as $\Delta B_\parallel$). The respective $g$-factors are also strongly anisotropic: $g_\perp$ is considerably shifted with respect to the free-electron value of $g_0 = 2.0023$, and is temperature-dependent, whereas $g_\parallel$ is barely shifted and is temperature independent[45, 46].

Although several explanations have been proposed[45–48], no consistent picture has emerged yet for these anomalous findings in graphite. Nevertheless, understanding the spin-relaxation mechanism would be important for the advancement of spin-relaxation theory in general, but especially for spintronics applications of mono- or few-layer graphene.

We unravel the anomalous spin relaxation in graphite by studying the details of the spin-orbit coupling, its dependence over the Fermi surface, and its anisotropy. We find that the SOC is strongly anisotropic in graphite due to the symmetry[47]: the (pseudospin) spin-orbit field is oriented along $z$. This results in a significant anisotropy of the electron spin dynamics, spin-relaxation time, and a giant $g$-factor anisotropy. We demonstrate that the ESR data is compatible with the SOC anisotropy scenario. Remarkably, the theory predicts an ultralong spin-relaxation time for spins that are polarized perpendicular to the graphite $(a, b)$ plane. Indeed, saturated ESR experiments reveal a spin-relaxation time longer than 100 ns *at room temperature* for this geometry. These findings qualify graphite thin film as a strong candidate for spintronics technology.

## Theoretical predictions

We performed first-principles calculations of the electronic structure of graphite in the presence of spin-orbit coupling. Figure 1. shows the band dispersions near the **K** point of the Brillouin zone (BZ). We limit our qualitative considerations to the shown energy scale, which corresponds to the quasiparticle energy smearing up to room temperature. The full calculated band structure along high symmetry lines is presented in the Supporting Information (SI).

In Elliott-Yafet's theory of spin relaxation[49, 50], which is dominant in centrosymmetric metals such as graphite, spin-relaxation is due to the mixing of the otherwise pure spin up/down states[9, 51–55]. For a selected spin quantization axis, the two degenerate Bloch states can be characterized as up, $|\uparrow\rangle$, or down, $|\downarrow\rangle$, in the absence of SOC. Due to spin-orbit coupling, the spin up/down states have a (typically small) admixture of the Pauli spin down/up spinors:

$$|\widetilde{\uparrow}\rangle_k = [a_k|\uparrow\rangle + b_k|\downarrow\rangle]e^{ikr}, \qquad (1)$$

$$|\widetilde{\downarrow}\rangle_k = [a_{-k}^*|\downarrow\rangle + b_{-k}^*|\uparrow\rangle]e^{ikr}. \qquad (2)$$

Here $|\widetilde{\uparrow}\rangle$, and $|\widetilde{\downarrow}\rangle$ are the Bloch states in the presence of the SOC. The spin-flip probability is proportional to $b_k^2$.

Thanks to our high precision calculation of the dispersion relation, we could determine $b_k^2$ near the Fermi level in Fig. 1. For the electron spin quantized along the $c$-axis, the spin admixture probability is rather small, less than $10^{-6}$, without a significant momentum dependence. However, $b_k^2$ exhibits peaks at the band crossings (in fact, those are anticrossings as SOC opens a gap of 24 µeV such as in graphene[56]) when the spin quantization axis is in the plane. Then the spin admixture probabilities are orders of magnitude higher than elsewhere in the Brillouin zone. In fact, we find here so-called spin hot-spots[51], similar to what happens in monolayer graphene at the Dirac point[57]. Figure 1 reveals that the dominant contribution to the spin admixture comes from the momenta along **K → Γ**.

While calculating Fermi-level averages of $b_k^2$ is beyond the scope of the present work (mainly due to the tiny Fermi surface of graphite and the presence of the spin hot-spots), our calculation suggests that the spin-relaxation rate in an in-plane magnetic field is expected to be at least an order of magnitude faster than in a magnetic field parallel to the $c$-axis.

To relay on the spin-relaxation rate to the ESR line, we evoke the theory of Yafet[50] developed for the case of an anisotropic SOC. For graphite, the major features could be nicely followed in Fig. 2. He argued that both the $T_1$ and $T_2$ relaxation times are caused by fluctuating magnetic fields $\delta \mathbf{B} = (\delta B_a, \delta B_b, \delta B_c)$ due to the SOC. Fluctuating fields along a given direction give rise to spin relaxation of a spin component perpendicular to them. (A more recent and rigorous derivation of the ESR relaxation times is found in ref. 58). Following Yafet[50], we introduce $\delta B_\perp^2$ and $\delta B_\parallel^2$ for the squared magnitude of the fluctuating fields along the crystalline $c$ axis, and in the $(a, b)$

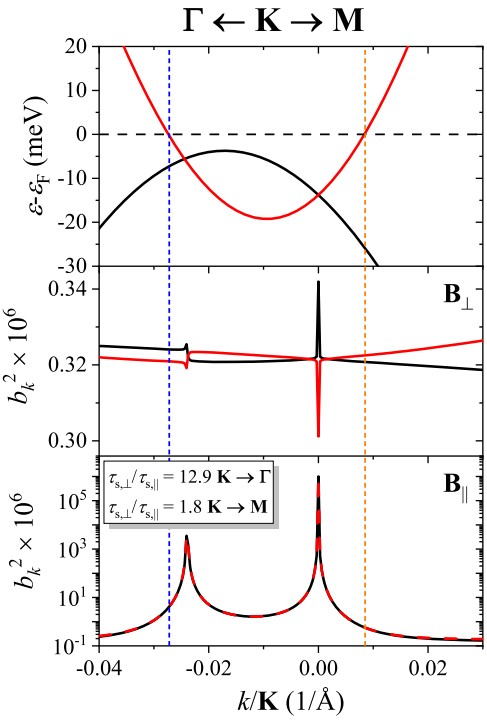

**Fig. 1 | Top: calculated low-energy electronic band structure of graphite at K along the lines towards Γ and M points.** Note the band degeneracy near the **K** point. Middle and bottom: spin-mixing parameters $b_k^2$ for the spin quantization (magnetic field **B**) along $c$ and in the plane $(a, b)$, respectively. Also indicated are the spin relaxation anisotropies for the momenta at the vertical dashed lines, implied by the calculated $b_k^2$ there.

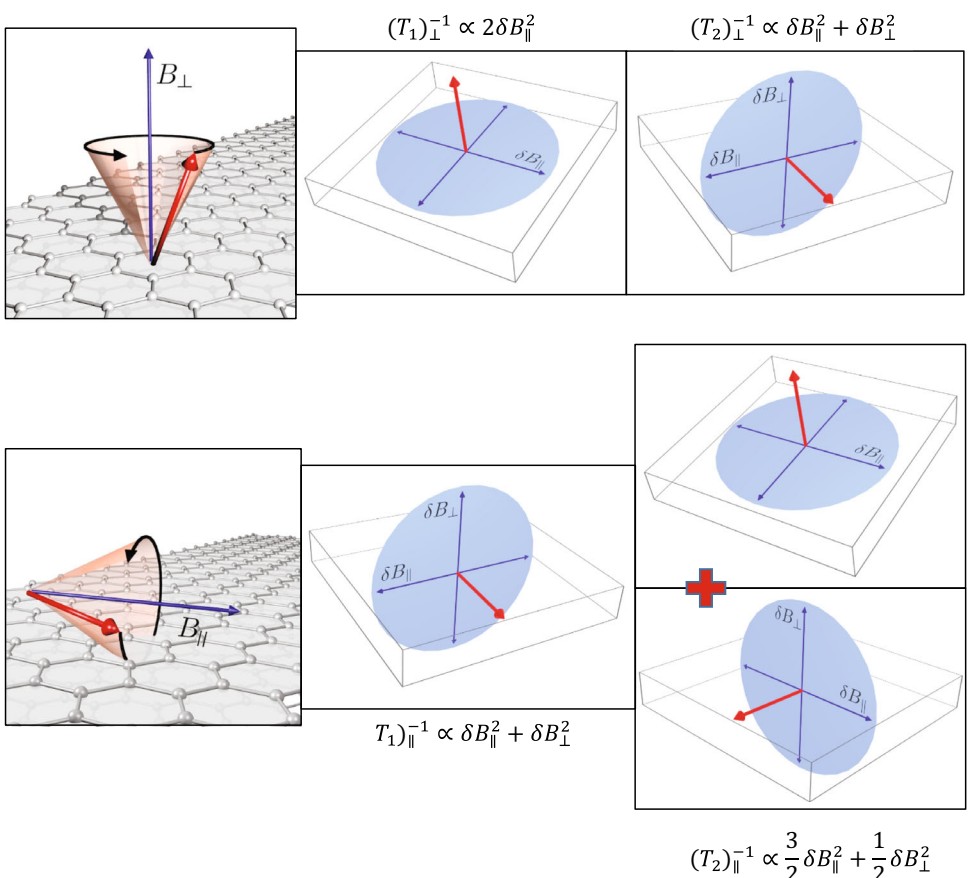

**Fig. 2 | Schematics of the geometry of the different spin-relaxation contributions for $B_\perp$ (upper panel) and $B_\parallel$ (lower panel).** Note that spins precess around the magnetic field for both orientations. The respective spin-relaxation rates are caused by the fluctuating components which are perpendicular to the given direction. For the $B_\parallel$, it consists of two contributions due to the Larmor precession of the spins.

plane, respectively. The results for the corresponding spin-relaxation rates are:

$$(T_1)^{-1}_\perp \propto 2\delta B^2_\parallel \quad (3)$$

$$(T_2)^{-1}_\perp \propto \delta B^2_\parallel + \delta B^2_\perp \quad (4)$$

$$(T_1)^{-1}_\parallel \propto \delta B^2_\parallel + \delta B^2_\perp \quad (5)$$

$$(T_2)^{-1}_\parallel \propto \frac{3}{2}\delta B^2_\parallel + \frac{1}{2}\delta B^2_\perp. \quad (6)$$

This is illustrated in Fig. 2. E.g., for $B_\perp$, spins precess around the $c$ axis and $T_1$ is caused entirely by $\delta B_\parallel$ (Eq. (3)), however $T_2$ is caused by both $\delta B_\parallel$ and $\delta B_\perp$ fluctuating fields as these SOC fields are perpendicular to spin component which is in the plane (Eq. (4)). For the $B_\parallel$ orientation, the spins precess in the $a-c$ plane thus $T_1$ is caused by both the $\delta B_\parallel$ and $\delta B_\perp$ which also yields $T_{2,\perp} = T_{1,\parallel}$ (Eqs. (4) and (5)). Yafet discussed the case of the extreme uniaxial anisotropy, i.e., when $\delta B^2_\parallel = 0$ while $\delta B^2_\perp$ is finite, which is in fact, due to symmetry considerations, the theoretical prediction for graphene (ref. 59). We suppose that this is the case for graphite, as well, which we check experimentally. It is interesting to notice, that somewhat counterintuitively, the extreme anisotropy corresponds to just a factor of 2 anisotropy in the ESR linewidth as $(T_2)^{-1}_\perp/(T_2)^{-1}_\parallel = \Delta B_\perp/\Delta B_\parallel = 2$ (see Eqs. (4) and (6)).

## Results and discussion

### Anisotropy of the ESR linewidth and *g*-factor

Figure 3 shows the temperature dependence of the ESR linewidth for the two major magnetic field orientations in highly-oriented pyrolytic graphite (HOPG) samples. Comparing the data with previous measurements[43, 46, 60–62], we find good agreement for the 50–300 K temperature range. Due to the finite penetration depth of microwaves, the ESR lineshape is asymmetric. This is taken into account by fitting the ESR spectra to the so-called Dysonian lines[63] that are well approximated by a mixture of absorption and dispersion components of a Lorentzian curve[64]. Additional data including typical ESR spectra, the *g*-factor, the temperature-dependent ESR intensity, angular dependence of the ESR spectra, and linewidth are provided in the SI.

Above 50 K, the linewidth increases with decreasing temperature for both orientations. Linewidth data for $B_\perp$ are not shown below 30 K where the significant line broadening and the relatively weak signal intensity prevent a reliable analysis. Annealing at 300 °C in a dynamic vacuum affects the ESR linewidth only below 50 K. This is plausible since due to its semi-metallic character and the small Fermi surface even a few ppm of paramagnetic impurities can dominate the signal at low temperatures due to their Curie spin susceptibility[65]. Higher

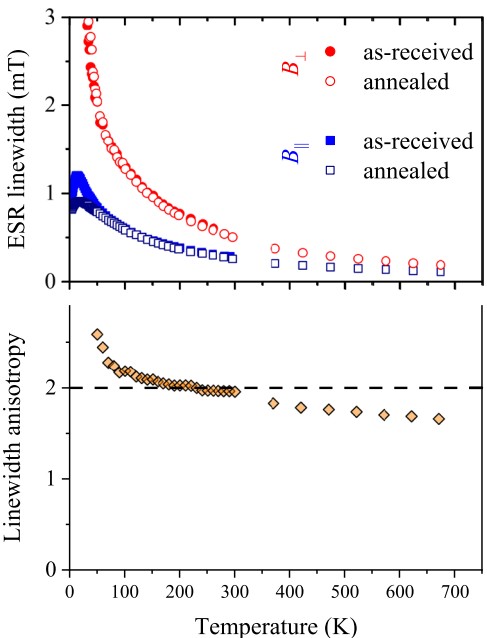

**Fig. 3 | Temperature-dependent ESR data at 9.4 GHz (~0.3 T) in graphite for both crystallographic orientations.** Upper panel: the ESR linewidth with different symbols for the as-obtained and annealed samples. For $B_\perp$, the data is shown only above 30 K. Lower panel: linewidth anisotropy factor, i.e. the ratio of the linewidth for $B_\perp$ and $B_\parallel$ configurations. The dashed line is the constant 2.

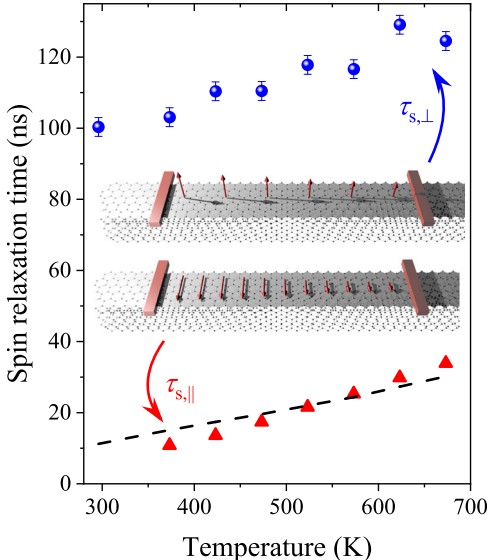

**Fig. 4 | The spin-lattice relaxation time, $T_1$, for the two orientations of the magnetic field (red triangles: $B_\parallel$, blue dots: $B_\perp$) in graphite as a function of temperature.** Note the about 10 times longer $T_1$ when $B_\perp$ as compared to $B_\parallel$. The inset depicts the expected experimental situation in a spin transport experiment. Dashed line denotes $T_2$ when $B_\perp$, which also confirms the case of extreme anisotropy as it aligns well with the $T_1$ values for $B_\parallel$, as predicted by Eqs. (4) and (5).

annealing temperatures (up to 600 °C) did not influence further the linewidth.

From a calibrated measurement of the spin-susceptibility, we obtained that the Curie contribution to the ESR signal is due to 5(1) ppm of $S = 1/2$ spins. We show in Fig. S2 that the susceptibility contribution from the localized and delocalized spins is equal at ~15 K. Given that any influence of the localized species on the $g$-factor and the linewidth is weighted with the spin-susceptibility[65], we conclude that above 50–100 K (especially at the technologically important room temperature region), the observed ESR linewidth is the intrinsic property of graphite. Although the present work focuses on the anisotropy of the ESR linewidth, we also mention that the conventional Elliott-Yafet's theory predicts an opposite temperature dependence, which may be related to the near band degeneracy around the graphite Fermi surface. Its full description requires additional work though.

The temperature-dependent $g$-factor data (see Fig. S5 of SI) confirms the earlier experimental data[46, 60–62] and the theoretical prediction. The measured factor of 2 anisotropy in the ESR linewidth of graphite (see the bottom image of Fig. 3) matches *exactly* the theoretical prediction of a vanishing SOC for spins polarized in the $(a, b)$ plane and a finite SOC for spins polarized along the $c$-axis. Strictly speaking, the linewidth anisotropy ratio is 2 only between 100 and 300 K, while it deviates somewhat upwards below 100 K and downwards above 300 K, which calls for more detailed modeling.

**The ultralong spin-relaxation time, $T_1$**

An important consequence of the absence of spin-orbit fields in the graphene planes is the predicted ultralong spin-lattice relaxation time, $T_{1,\perp}$, according to Eq. (3), whereas the other three relaxation times remain finite, below 50 ns. To experimentally verify this prediction, we performed saturation ESR measurements (see Fig. S7 of SI). Although the transversal spin-relaxation time, $T_2$, is accessible directly from the linewidth, $T_1$ does not directly affect the lineshape in conventional ESR studies. In principle, both relaxation times can be measured with a

spin-echo technique but it is limited to relaxation times when both $T_1$ and $T_2$ are longer than ~1 μs, which is not the case herein. However, saturation ESR experiments[44] yield relaxation times down to ~10 ns. The method is based on monitoring the variation of the ESR signal intensity and the linewidth as a function of the irradiating microwave power. A characteristic drop in the signal intensity, accompanied by a line broadening allows determining $T_1$. Further technical details are given in the SI together with the raw saturation ESR data.

The $T_1$ results for both orientations of the magnetic field are shown in Fig. 4. The data is given only above 300 K for $B_\perp$ and above 370 K for $B_\parallel$ as below these temperatures the shortening of the respective $T_2$ prevents the observation of the saturation effect. Nevertheless, having in mind the spintronics application, this is the relevant temperature range. Red triangles and the dashed line in Fig. 4 confirm that the $T_{1,\parallel} = T_{2,\perp}$ prediction from Eqs. (4) and (5) is indeed satisfied, which provides further proof that the extreme SOC anisotropy is realized in graphite.

The most striking experimental observation is the presence of $T_{1,\perp}$ relaxation times beyond 100 ns and the approximate factor 10 anisotropy of $T_1$. Although it is observed in an ESR experiment, i.e., in the presence of an external magnetic field, this result can be readily extended to the case of zero magnetic fields: it implies that $\tau_{s,\perp}$ in graphite would be as long as 100 ns when electrons are injected with a spin perpendicular to the graphene planes and it would be about a factor 10 times shorter when the spins lie in the graphene planes. This is also depicted in Fig. 4. The giant anisotropy may find a number of applications, e.g., it could be exploited to control the spin-relaxation times in spintronic devices.

It is known[1] that electrons travel mainly in the graphene planes in graphite and interlayer transport is diffusion limited and it leads to a conduction anisotropy $\sigma_\parallel/\sigma_\perp$ beyond $10^3$–$10^4$. This allows to approximate the spin diffusion in graphite while considering that electrons reside on a given graphene layer. The large value of $\tau_{s,\perp}$ leads to an ultra-long spin-diffusion length in graphite: $\delta_s = \frac{v_F}{\sqrt{2}}\sqrt{\tau \tau_s}$ (the factor 2 appears from the two-dimensional diffusion equation). It gives $\delta_s \approx 70$ μm with typical values of $v_F = 10^6$ m/s and $\tau = 10^{-13}$ s. This is

already a macroscopic length scale, which could bring spintronic devices closer to reality.

## Summary

In conclusion, we unraveled the anomalous dynamics of itinerant electron spins in graphite. We first studied the details of the band-structure-related spin–orbit coupling which predicted a hitherto hidden, symmetry-related extreme anisotropy of the spin-orbit coupling. This anisotropy in fact explains the known anomalous anisotropic properties of graphite: a strongly anisotropic $g$-factor and the 1:2 anisotropy of the ESR linewidth. We recognized that the latter should in principle be accompanied by a giant anisotropy of the spin-lattice relaxation time. This prediction was examined with saturation ESR studies and we observe ultralong ($T_1$ in excess of 100 ns) spin-relaxation times for spins aligned perpendicular to the graphene planes. When extrapolated to the zero-field limit, this predicts similarly long spin-relaxation times in spin-transport studies and a macroscopic spin-diffusion length.

## Methods

### Experiment

We studied high-quality HOPG (highly-oriented pyrolitic graphite) from Structure Probe Inc. (SPI Grade I) using electron spin resonance (ESR). The HOPG had a mosaicity of $0.4° \pm 0.1°$. We studied X-band (0.33 T, 9.4 GHz) in a commercial ESR spectrometer (Bruker Elexsys E500) in the 4–673 K temperature range. The spectrometer is equipped with a goniometer which allows reliable and reproducible sample rotations. We used HOPG disk samples of 3 mm diameter and 70 μm thickness.

The samples were sealed in quartz ampules under 20 mbar He for the ESR measurement. Annealing of some of the samples was performed in a high dynamic vacuum in a furnace up to 300–600 °C. The goals in this study are the accurate measurement of the ESR linewidth, and temperature-dependent ESR intensity (penetration effects for a bulky sample prevent the determination of the absolute spin-susceptibility). We also monitored the $g$-factor as a function of temperature to enable comparison with existing literature data. We employed a low magnetic field modulation to avoid signal distortion, we also used a low (150 μW) microwave power to avoid saturation of the normal data and we monitored the change in the cavity resonance frequency. We quote the widths of derivative Lorentzian curves (or half width-half maximum data, HWHM) fitted to the experimental data which is related to the peak-to-peak linewidth as: $\Delta B_{HWHM} = \Delta B_{pp}/\sqrt{3}$. Saturated ESR experiments were performed in a commercial microwave cavity (Bruker ER 4122 SHQ, Super High $Q$ Resonator) with an unloaded $Q_0 = 7500$ and with the sample $Q_L = 5500$. This cavity produces $AC$ magnetic fields of $B_1 = 0.2$ mT $\sqrt{\frac{pQ_L}{Q_0}}$, where $p$ is the microwave power in Watts and we used microwave powers up to 0.2 W.

### Theory

The electronic band structure of graphite in the presence of spin-orbit coupling is determined by the full potential linearized augmented plane waves (LAPW) method, based on density functional theory implemented in Wien2k[66]. For exchange-correlation effects, the generalized gradient approximation was utilized[67]. In our three-dimensional calculation the graphene sheets of lattice constant $a = 1.42\sqrt{3}$ Å are separated by the distance of $c = 3.35$ Å. Integration in the reciprocal space was performed by the modified Blöchl tetrahedron scheme, taking the mesh of $33 \times 33$ $k$-points in the irreducible Brillouin zone (BZ) wedge. As the plane-wave cut-off, we took 9.87 Å$^{-1}$. The 1s core states were treated fully relativistically by solving the Dirac equation, while spin-orbit coupling for the valence electrons was treated within the muffin-tin radius of 1.34 a.u. by the second variational method[68].

The algorithm and the code used to calculate the $b_k^2$ values were proven to be correct for various other materials previously[69]. These include WS$_2$[70], WSe$_2$, MoSe$_2$[71], various heterostructures[72,73], etc. We are confident that it provides adequate results for the case of graphite as well. However, averaging the spin admixture, $b^2$, over the tiny Fermi-surface pockets is, at the moment, not feasible. The main reason is that the admixture varies strongly (over several orders of magnitude) close to the band degeneracies (spin hot spots), requiring very dense sampling of the Fermi surface. At the moment, this is beyond the computational capabilities of our available computing infrastructure.

## Data availability

The data needed to evaluate and reproduce the conclusions are present in the paper and the Supporting Information (Online Content). Additional data related to this paper are available from the corresponding author upon request.

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

## Acknowledgements

Work supported by the National Research, Development and Innovation Office of Hungary (NKFIH) Grants Nr. K137852, K142179, 2022-2.1.1-NL-2022-00004, and 2019-2.1.7-ERA-NET-2021-00028. The Swiss National Science Foundation (Grant No. 200021 144419), DFG (German Research Foundation), SFB 1277 (Project No. 314695032), SPP 2244 (Project No. 443416183), European Union Horizon 2020 Research and Innovation Program under contract number 881603 (Graphene Flagship), and the project FLAG ERA JTC 2021 2DSO-TECH are acknowledged. M.G. acknowledges financial support from Slovak Research and Development Agency provided under Contract No. APVV-SK-CZ-RD-21-0114 and by the Ministry of Education, Science, Research and Sport of the Slovak Republic provided under Grant No. VEGA 1/0105/20 and Slovak Academy of Sciences project IMPULZ IM-2021-42.

## Author contributions

B.G.M., P.S., B.N., and T.F. performed the ESR studies under the supervision of L.F.; B.G.M. performed the high temperature saturated ESR studies. M.G., V.Z., and J.F. performed the band structure calculations and the analysis of the SOC. B.D. and F.S. developed the model for the temperature-dependent ESR linewidth and $g$-factor. G.C. performed numerical simulations of the ESR lineshape. F.S. and J.F. outlined the overall explanation for the spin-relaxation properties. All authors contributed to the writing of the manuscript.

## Competing interests

The authors declare no competing interests.
