## [Peer Review File · Nature Communications]

Reviewers' Comments:

Reviewer #2:

Remarks to the Author:

The manuscript by Markus and colleagues reports on the theory and experimental determination of the spin relaxation time T_1 in graphite. The authors find that T_1 reaches 120 ns near room temperature when the magnetic field points along the c axis, thus perpendicular to the graphene layers. This result is certainly interesting and quite intriguing, as the measured T_1 largely exceeds values found previously for graphene and because the spin relaxation plays a crucial role on the potential of C-based semiconductors for spintronics. Yet, the interpretation given remains qualitative, only accounts for some of the observed features and does not seem very original. Therefore, my opinion is that the manuscript, as it is, does not provide a sufficiently convincing new evidence to justify its publication in such a high profile journal as Nature Communications. Some further comments/questions and suggestions for improvement follow next.

1. Experiments:

o The anisotropy observed in the effective electronic g -factor and in the linewidth have been reported previously, with qualitatively similar results, as the authors acknowledge in the text. A curiosity here is that the linewidth data shown in Fig. 3 do not agree with those reported in Fig. 1 and Fig. 3 from ref [46]. There seems to be a factor 2 between both sets of data, which might deserve a comment.

o Besides the contribution from the graphite electrons, the ESR spectra show additional signals likely arising from magnetic impurities. Since these localized spins might have some influence on T_1 and T_2 (see ref [[17]]), it would be worth trying to quantify their concentration. When the magnetic field points within the graphene layers, the resonant field of this signal nearly coincides with that of the electrons (Fig. S2), but in the perpendicular orientation (Fig. S6) the two are clearly separated. Incidentally, I wonder if in this case the electronic intensity follows the expected temperature dependence associated with the change in the density of carriers.

o The main original result of this work is the determination of T_1 from the power dependence of ESR spectra. I wonder if the existence of some inhomogeneous broadening, e.g. arising from interactions with the impurity spins, might introduce some error in the determination of T_1 (as it depends on a previous knowledge of T_2).

2. Theory:

o The theory is based mainly on the Elliot-Yafet mechanism (reference [[50]]) that arises from the interaction of spins with phonons mediated by the spin-orbit interaction. However, the information extracted from these calculations is a bit deceptive, as the main "prediction" is that given by Eqs. (3-6), which mainly reflect geometric factors without really providing any hint about the values of the effective field fluctuations that are responsible for the spin relaxation. This is still a long way from the "full description of the relaxation times" promised at the paper's abstract. The agreement with some observed features might therefore be a bit misleading. Although the anisotropy in g and ΔH are indeed accounted for by these expressions, the main discussion and interest about T_1 focusses on the large quantitative discrepancies found between the values expected/hoped for C-based materials (longer than 1 micro-second) and those actually measured. Besides, one expects that the theory describes at least the temperature dependence of the relaxation time (see point 3 below). I strongly suggest the authors exploit their knowledge about the electronic structure in graphite for different magnetic field orientations (Fig. 1) to derive expressions for T_1 and T_2 (say, following the procedure that was described in ref[46] for T_2). The discussion following Eqs. (1) and (2) mentions that T_1 "in an in-plane magnetic field is expected to be about 10 times faster than in a magnetic field parallel to the c axis" but this is not substantiated by actual calculations or detailed formulas.

3. Interpretation and impact of the results:

o Spin relaxation mechanism: The conclusions suggest that the long spin relaxation times measured with a magnetic field applied along the c axis of graphite are a consequence of the strong anisotropy of the spin-orbit interaction, thus are associated to the intrinsic relaxation mechanism described by Elliot and Yafet. That this anisotropy should lead to a very long relaxation time for spins pointing perpendicular to the graphene layers is not new (see e.g. [46]). In order to actually show that this mechanism is dominant, I feel that additional evidences are necessary. As I mentioned above, one should actually compare the predicted and measured values quantitatively. Although 120 ns is indeed quite long, it might be that it remains below that expected for the

intrinsic relaxation processes, as it happens in the case of graphene. Also, it would be just fair to explore other possible scenarios, e.g. relaxation induced by resonant coupling to impurity spins. These spins are indeed present in the samples (see Figs. S1 and S6) and, as shown in ref [17], they can provide a (very) efficient relaxation channel. Discussing the results in connection with this alternative theory, which could be backed by estimations of the impurity spin concentrations from ESR data, would help much to find a suitable and robust interpretation of the results.

o Temperature dependence of T1: The dependence of g and ΔH on temperature that is shown in Fig. 3 seems to be compatible with a mechanism based on the anisotropic spin-orbit coupling, as was shown already in ref [46]. However, the decrease of T1 with temperature I found a bit surprising. The original calculations by Yafet lead to a linear, or even stronger at lower T, decrease of $1/T_1$ (not T1) with T. I think this also deserves a comment.

o Impact: The extrapolation of the results obtained by ESR to the conditions relevant to spintronic devices (weak or zero fields) is not straightforward. It depends on the dominant relaxation mechanism. For instance, if impurity spin fluctuations play a role, one could expect that T1 decreases with decreasing magnetic field. Also, I miss the connection with graphene, which is mentioned at the introduction as one of the motivations of this work. What makes graphite different from graphene that precludes such long T1 values to be observed in the latter? This also needs to be discussed, I think.

Reviewer #3:

Remarks to the Author:

The authors report experimental data of T1 and T2 spin relaxation times in graphite. The results are accompanied by detailed theoretical results of the band structure. The manuscript is potentially of interest for a wide audience given the continued interest in using graphene (graphite for that matter) for spintronic applications.

I am not qualified to question the quality of the experimental results. I am assuming that the authors have made a significant advancement in the methodological area to measure T1 for the first time, as they claim. I only have one comment. In Fig. 4 they show relaxation times which are supposed to correlate with those measured in a different set of experiments (transport experiments). There is a plethora of these in the literature. How do the results compare?

Regarding the theory I have a couple of comments. 1) I do not see a significant advancement in the evaluation of the spin mixing b_k coefficients compared to many other works of the authors. Can they point out what they mean by results of "great confidence"? 2) How do they obtain the ratio $\tau_{\text{perp}}/\tau_{\text{par}}$ for the different paths K-Gamma and K-M shown in the legend of Fig. 1? Is it an average?. 3) I would have liked to see a connection between these calculations and the phenomenological theory of Yafet, which they use to explain the experimental results. How can one relate the "fluctuating fields" with the b_k coefficients? Otherwise these two pieces of theory are a bit disconnected, the second one not being particularly supported by the calculations.

Response to the Reviewers

We gratefully thank both reviewers for the thorough reviews and for the important and useful remarks and observations helping us to resubmit an improved version of our MS.

Quote from the report is italic

Our response is normal

Changes made to the text are in red and bold

Reviewer #2:

The manuscript by Markus and colleagues reports on the theory and experimental determination of the spin relaxation time T_1 in graphite. The authors find that T_1 reaches 120 ns near room temperature when the magnetic field points along the c axis, thus perpendicular to the graphene layers. This result is certainly interesting and quite intriguing, as the measured T_1 largely exceeds values found previously for graphene and because the spin relaxation plays a crucial role on the potential of C-based semiconductors for spintronics. Yet, the interpretation given remains qualitative, only accounts for some of the observed features and does not seem very original. Therefore, my opinion is that the manuscript, as it is, does not provide a sufficiently convincing new evidence to justify its publication in such a high profile journal as Nature Communications. Some further comments/questions and suggestions for improvement follow next.

We thank the Referee for the thorough review and the remarks that allowed us to improve the quality of the manuscript. We really appreciate his/her efforts and hope that with our answers and corrections, we can convince the Referee that the MS is suitable for Nature Communications.

1. Experiments:

- The anisotropy observed in the effective electronic g -factor and in the linewidth have been reported previously, with qualitatively similar results, as the authors acknowledge in the text. A curiosity here is that the linewidth data shown in Fig. 3 do not agree with those reported in Fig. 1 and Fig. 3 from ref [46]. There seems to be a factor 2 between both sets of data, which might deserve a comment.

We thank the reviewer for this comment. The difference between the two datasets is arising from the fact that Hueber *et al.* [46] use the peak-to-peak linewidth definition, while we use the HWHM of the fitted curve as ΔB . The scaling factor between the two is $\Delta B_{\text{HWHM}} = \Delta B_{\text{pp}}/\sqrt{3}$. After rescaling, the two datasets match perfectly with each other. We added a note in the text (Methods section, after “we monitored the change in the cavity resonance frequency.”): **“We quote the widths of derivative Lorentzian curves (or half width-half maximum data, HWHM) fitted to the experimental data which is related to the peak-to-peak linewidth as: $\Delta B_{\text{HWHM}} = \Delta B_{\text{pp}}/\sqrt{3}$.”**

- Besides the contribution from the graphite electrons, the ESR spectra show additional signals likely arising from magnetic impurities. Since these localized spins might have some influence on T_1 and T_2 (see ref [[17]), it would be worth trying to quantify their concentration. When the magnetic field points within the graphene layers, the resonant field of this signal nearly coincides with that of the electrons (Fig. S2), but in the perpendicular orientation (Fig. S6) the

two are clearly separated. Incidentally, I wonder if in this case the electronic intensity follows the expected temperature dependence associated with the change in the density of carriers.

We have used similar graphite samples as described in Ref. [72] and Ref. [73]. The authors in both papers conclude that in graphite paramagnetic iron is the most common impurity. According to the cited articles, in the highest quality HOPG samples (Grade I) their concentration is ~ 1 -5 ppm, and ~ 5 -10 ppm in mid-grade ones (Grade III). We have evaluated the number of localized spins (it is about 5 ppm) in our sample. Concerning the second part of the question, the observed ESR intensity is indeed flat above 50 K, which confirms that itinerant electron spins dominate the signal. Although the localized spins can affect the measurable properties (g -factor and the linewidth), their influence scales with the spin-susceptibility (Ref. [65], Barnes Adv. Phys. 1980). This also means that above about 50-100 K, their influence is negligible. This is supported by the observed flat susceptibility above 50 K and is in good agreement with previous literature data at various field strengths. To clarify these points, we replaced the original statement: “It is an important fact that above 50 K (especially at the technologically important room temperature region), the observed ESR linewidths are the intrinsic property of HOPG” by **“From a calibrated measurement of the spin-susceptibility, we obtained that the Curie contribution to the ESR signal is due to 5(1) ppm of $S=1/2$ spins. We show in Fig. S2 that the susceptibility contribution from the localized and delocalized spins is equal at about 15 K. Given that any influence of the localized species on the g -factor and the linewidth is weighted with the spin-susceptibility [65], we conclude that above 50-100 K (especially at the technologically important room temperature region), the observed ESR linewidth is the intrinsic property of graphite.”**

- The main original result of this work is the determination of T_1 from the power dependence of ESR spectra. I wonder if the existence of some inhomogeneous broadening, e.g. arising from interactions with the impurity spins, might introduce some error in the determination of T_1 (as it depends on a previous knowledge of T_2).

As we discussed in the previous point, the impurity spins have a negligible effect on *any* measurable properties at room temperature. Any influence scales with spin-susceptibility according to Ref. [65]. S2 in the Supplementary information shows that the two contributions are equal at around 15 K therefore, at room temperature the spin-susceptibility from the localized spins is about 20 times smaller than that of the itinerant electrons in graphite. This argument is supported by previous literature data: were any significant inhomogeneous broadening present, the linewidth measured at 35 GHz would significantly deviate from the X-band data, which is not the case [46].

2. Theory:

- The theory is based mainly on the Elliot-Yafet mechanism (reference [[50]]) that arises from the interaction of spins with phonons mediated by the spin-orbit interaction. However, the information extracted from these calculations is a bit deceptive, as the main “prediction” is that given by Eqs. (3-6), which mainly reflect geometric factors without really providing any hint about the values of the effective field fluctuations that are responsible for the spin relaxation. This is still a long way from the “full description of the relaxation times” promised at the paper’s abstract. The agreement with some observed features might therefore be a bit misleading. Although the anisotropy in g and ΔH are indeed accounted for by these expressions, the main discussion and interest about T_1 focusses on the large quantitative

discrepancies found between the values expected/hoped for C-based materials (longer than 1 micro-second) and those actually measured. Besides, one expects that the theory describes at least the temperature dependence of the relaxation time (see point 3 below). I strongly suggest the authors exploit their knowledge about the electronic structure in graphite for different magnetic field orientations (Fig. 1) to derive expressions for T_1 and T_2 (say, following the procedure that was described in ref[46] for T_2). The discussion following Eqs. (1) and (2) mentions that T_1 “in an in-plane magnetic field is expected to be about 10 times faster than in a magnetic field parallel to the c axis” but this is not substantiated by actual calculations or detailed formulas.

Concerning the first part of the question: we accept the criticism and change the phrase: “we give a full description of the relaxation times. We have measured the T_1 in graphite by saturation ESR measurements and find it markedly different from T_2 .” in the abstract to “**we predict an unexpected behavior of the relaxation times. We indeed find it experimentally from saturation ESR measurements that T_1 is markedly different from T_2 , which is accompanied by a peculiar anisotropy.**”

The second part of the suggestion, i.e., to provide a comprehensive description of the full temperature dependence, is unfortunately beyond our scope. The peculiar anisotropy and inequality of the two relaxation times are already astonishing. Its description of a band structure-based model is in our opinion reassuring enough, especially when the theoretical prediction is supported by clear-cut experiments. We bring attention to the fact that the result of this peculiar anisotropy of spin relaxation in graphite has been observed for more than half a century, but its true origin has not been clarified, which in fact is presented herein.

Concerning the third part of the question, we modify the phrase in question “is expected to be about 10 times faster” to “is expected to be **at least an order of magnitude** faster”, as it better explains the statement. The values in Fig. 1 hint at such a value although we did not provide an actual Fermi surface average for the spin admixture parameter. Therefore, we agree with the Referee that a less accurate statement on the difference is required.

3. Interpretation and impact of the results: We split the question into two parts.

- Spin relaxation mechanism: The conclusions suggest that the long spin relaxation times measured with a magnetic field applied along the c axis of graphite are a consequence of the strong anisotropy of the spin-orbit interaction, thus are associated to the intrinsic relaxation mechanism described by Elliot and Yafet. That this anisotropy should lead to a very long relaxation time for spins pointing perpendicular to the graphene layers is not new (see e.g. [46]). In order to actually show that this mechanism is dominant, I feel that additional evidences are necessary. As I mentioned above, one should actually compare the predicted and measured values quantitatively. Although 120 ns is indeed quite long, it might be that it remains below that expected for the intrinsic relaxation processes, as it happens in the case of graphene.

We would like to reformulate the Referee’s statement: we not only speculate on the possibility of a very long T_1 for $B||c$, but we explicitly *measure* it. It is indeed the consequence of the peculiar anisotropy of the SOC, which in our opinion has remained unexplained in the literature. We have not seen a detailed discussion such as that presented in Fig. 2 and Equations (3)-(6) which undoubtedly explain the observed linewidth anisotropy with the SOC anisotropy. Our description, including that of the g -factor, is self-consistent, which is the additional evidence

that the Referee lacks. 120 ns is indeed about 1-2 orders of magnitude longer than that expected or predicted. However, as we mention in our manuscript, one does not expect a truly *infinitely* long T_1 , as at a certain point other competing mechanisms, such as e.g., the aforementioned weak but non-zero relaxation due to impurities or higher-order (e.g., Orbach-like) relaxation processes may start to dominate in the beyond-100 ns T_1 regime.

Also, it would be just fair to explore other possible scenarios, e.g. relaxation induced by resonant coupling to impurity spins. These spins are indeed present in the samples (see Figs. S1 and S6) and, as shown in ref [17], they can provide a (very) efficient relaxation channel. Discussing the results in connection with this alternative theory, which could be backed by estimations of the impurity spin concentrations from ESR data, would help much to find a suitable and robust interpretation of the results.

Our manuscript devotes a great deal to proving that the observed effect is indeed intrinsic to graphite and that the role of impurities can be excluded as it is discussed above. Besides the above-mentioned estimates on the number of impurities as well as the role they may play on the spin relaxation properties, we mention that

- the temperature dependence of the spin susceptibility above 50 K
- the accurate reproducibility of the g-factor and linewidth data for several measurements by various groups and on various samples during the past 70 years

unambiguously excludes the possibility raised by the Referee.

- Temperature dependence of T_1 : The dependence of g and ΔH on temperature that is shown in Fig. 3 seems to be compatible with a mechanism based on the anisotropic spin-orbit coupling, as was shown already in ref [46]. However, the decrease of T_1 with temperature I found a bit surprising. The original calculations by Yafet lead to a linear, or even stronger at lower T, decrease of $1/T_1$ (not T_1) with T. I think this also deserves a comment.

We thank the Referee for these suggestions. We added in the Results and Discussion, before the paragraph “The temperature-dependent g-factor”, the following sentence: “**Although the present work focuses on the anisotropy of the ESR linewidth, we also mention that the conventional Elliott-Yafet’s theory predicts an opposite temperature dependence, which may be related to the near band degeneracy around the graphite Fermi surface. Its full description requires additional work though.**”

- Impact: The extrapolation of the results obtained by ESR to the conditions relevant to spintronic devices (weak or zero fields) is not straightforward. It depends on the dominant relaxation mechanism. For instance, if impurity spin fluctuations play a role, one could expect that T_1 decreases with decreasing magnetic field. Also, I miss the connection with graphene, which is mentioned at the introduction as one of the motivations of this work. What makes graphite different from graphene that precludes such long T_1 values to be observed in the latter? This also needs to be discussed, I think.

As mentioned above, the dominance of the impurity spins on the spin relaxation can be excluded with confidence. Graphite is clearly related to graphene, being its three-dimensional counterpart, therefore using the intensive research on graphene as a motivation for our work is warranted. Intrinsically, graphene could and should behave similarly to graphite as far as the spin-relaxation properties are concerned, but it being a single layer, it is more prone to extrinsic

effects and is more affected by its local environment, while graphite is a bulk material and more resilient to these.

Reviewer

#3:

The authors report experimental data of T_1 and T_2 spin relaxation times in graphite. The results are accompanied by detailed theoretical results of the band structure. The manuscript is potentially of interest for a wide audience given the continued interest in using graphene (graphite for that matter) for spintronic applications.

We are grateful to the Reviewer for her/his efforts and for the kind judgment about our work. We believe that her/his suggestions contributed to improve the manuscript.

I am not qualified to question the quality of the experimental results. I am assuming that the authors have made a significant advancement in the methodological area to measure T_1 for the first time, as they claim. I only have one comment. In Fig. 4 they show relaxation times which are supposed to correlate with those measured in a different set of experiments (transport experiments). There is a plethora of these in the literature. How do the results compare?

As far as we know, there are no available spin transport measurements on graphite. In graphene, being the closest compound, the observed world record is 12.6 ns. This yields a factor of ten difference in favor of our observations.

Regarding the theory I have a couple of comments.

1) I do not see a significant advancement in the evaluation of the spin mixing b_k coefficients compared to many other works of the authors. Can they point out what they mean by results of "great confidence"?

We agree with the Referee that we may have overstated our claims. It suffices to say that the recent improvement of computer codes to the calculator of the spin-admixture parameters is now capable of calculating these. The phrase "great confidence" is thus deleted.

2) How do they obtain the ratio $\tau_{\text{perp}}/\tau_{\text{par}}$ for the different paths K-Gamma and K-M shown in the legend of Fig. 1? Is it an average?

The ratio in question is taken at k -points indicated by the dashed lines. It is not an average value; the Fermi energy crosses the electronic band at these quasi-momentum points. Therefore, these are the relevant points on the Fermi surface. Certainly, some level of averaging may be considered, either due to temperature or due to some uncertainty in the determination of the exact location of the FS.

3) I would have liked to see a connection between these calculations and the phenomenological theory of Yafet, which they use to explain the experimental results. How can one relate the "fluctuating fields" with the b_k coefficients? Otherwise these two pieces of theory are a bit disconnected, the second one not being particularly supported by the calculations.

This comment of the Referee is definitely correct, i.e., that the relation between the two kinds of descriptions is not straightforward. One should keep in mind that the two kinds of descriptions serve different purposes. The fluctuating model is very useful when describing the anisotropic effect and the different contributions to the anisotropic spin-relaxation times (T_1 versus T_2). In principle, the conventional Elliott-Yafet's, which uses the b_k coefficients, is incapable of explaining T_1 and T_2 separately, as it is a zero-field model and does not include the spin precession, which in turn is an important ingredient when explaining the different

behaviors of T_1 and T_2 . Rather, the b_k coefficients have a predicting value i.e. their calculation predicts the peculiar anisotropy in this material.

List of Changes

In the Abstract: “we give a full description of the relaxation times. We have measured the T_1 in graphite by saturation ESR measurements and find it markedly different from T_2 .”

is replaced by

“we predict an unexpected behavior of the relaxation times. We indeed find it experimentally from saturation ESR measurements that T_1 is markedly different from T_2 , which is accompanied by a peculiar anisotropy.”

the phrase “great confidence” is deleted

In the Section Theoretical Predictions “is expected to be about 10 times faster”

is replaced by

“is expected to be **at least an order of magnitude** faster”

A new paragraph is added to the Results and Discussion section:

From a calibrated measurement of the spin-susceptibility, we obtained that the Curie contribution to the ESR signal is due to 5(1) ppm of $S=1/2$ spins. We show in Fig. S2 that the susceptibility contribution from the localized and delocalized spins is equal at about 15 K. Given that any influence of the localized species on the g -factor and the linewidth is weighted with the spin-susceptibility [65], we conclude that above 50-100 K (especially at the technologically important room temperature region), the observed ESR linewidth is the intrinsic property of graphite. Although the present work focuses on the anisotropy of the ESR linewidth, we also mention that the conventional Elliott-Yafet’s theory predicts an opposite temperature dependence, which may be related to the near band degeneracy around the graphite Fermi surface. Its full description requires additional work though.

Methods section, after “we monitored the change in the cavity resonance frequency.” a sentence is added: **“We quote the widths of derivative Lorentzian curves (or half width-half maximum data, HWHM) fitted to the experimental data which is related to the peak-to-peak linewidth as: $\Delta B_{\text{HWHM}} = \Delta B_{\text{pp}}/\sqrt{3}$.”**

Reviewers' Comments:

Reviewer #2:

Remarks to the Author:

In my first report I expressed some concerns, related mainly to the influence that magnetic impurities might have on the results and on the interpretation. The author responses show that the impurities cannot play a major role in the temperature range in which the relaxation times and their anisotropies have been measured. The text also discusses these aspects quite clearly. The answers to other criticisms are also quite convincing. Even without a full fledged interpretation, I think the experimental results are important and can have a large impact. Therefore, I'm happy to recommend this manuscript for publication.

Reviewer #3:

Remarks to the Author:

I am not particularly satisfied with the way the authors have addressed my points. Saying "It suffices to say that the recent improvement of computer codes to the calculator of the spin-admixture parameters is now capable of calculating these." is a little disappointing. What improvements are we talking about? Are these so technical that the authors cannot spare a few lines in the supplementary material explaining them?

Anyway, I am aware that my concerns on the theory part are not strong enough as to invalidate the value of the experimental work as a whole and I recommend publication.

Response to the Reviewers

We gratefully thank both reviewers for the thorough reviews and for the important and useful remarks and observations helping us to resubmit an improved version of our MS.

Quote from the report is italic

Our response is normal

Changes made to the text are in red and bold

Reviewer #2:

In my first report I expressed some concerns, related mainly to the influence that magnetic impurities might have on the results and on the interpretation. The author responses show that the impurities cannot play a major role in the temperature range in which the relaxation times and their anisotropies have been measured. The text also discusses these aspects quite clearly. The answers to other criticisms are also quite convincing. Even without a full fledged interpretation, I think the experimental results are important and can have a large impact. Therefore, I'm happy to recommend this manuscript for publication.

We thank the Referee for the previous comments and suggestions. We are really grateful for recommending the MS for publication.

Reviewer #3:

I am not particularly satisfied with the way the authors have addressed my points. Saying "It suffices to say that the recent improvement of computer codes to the calculator of the spin-admixture parameters is now capable of calculating these." is a little disappointing. What improvements are we talking about? Are these so technical that the authors cannot spare a few lines in the supplementary material explaining them?

Anyway, I am aware that my concerns on the theory part are not strong enough as to invalidate the value of the experimental work as a whole and I recommend publication.

We are glad that the reviewer recommends our manuscript for publication. We are extremely sorry that we have formulated our previous response to the corresponding theory point in somewhat unclear terms. Our code is capable of calculating the b_k^2 values with the required precision. This was developed and probed previously for many cases but was never applied to the case of graphite before. Since the algorithm has been proven for many other materials, we believe the case holds for graphite, as well. However, calculating the b^2 average over the Fermi surface is beyond the limit of current computational techniques for graphite because of the peculiar shape of the FS. We have added the following sentences for clarification the end of the Methods section:

The algorithm and the code used to calculate the b_k^2 values were proven to be correct for various other materials previously⁶⁹. These include WS₂⁷⁰, WSe₂, MoSe₂⁷¹, various heterostructures^{72,73}, etc. We are confident that it provides adequate results for the case of graphite as well. However, averaging the spin admixture, b^2 , over the tiny Fermi-surface pockets is, at the moment, not feasible. The main reason is that the admixture varies strongly (over several orders of magnitude) close to the band degeneracies (spin hot spots), requiring very dense sampling of the Fermi surface. At the moment, this is beyond the computational capabilities of our available computing infrastructure.

List of Changes

We added the following sentences to the Methods section:

“The algorithm and the code used to calculate the b_{k^2} values were proven to be correct for various other materials previously⁶⁹. These include WS₂⁷⁰, WSe₂, MoSe₂⁷¹, various heterostructures^{72,73}, etc. We are confident that it provides adequate results for the case of graphite as well. However, averaging the spin admixture, b^2 , over the tiny Fermi-surface pockets is, at the moment, not feasible. The main reason is that the admixture varies strongly (over several orders of magnitude) close to the band degeneracies (spin hot spots), requiring very dense sampling of the Fermi surface. At the moment, this is beyond the computational capabilities of our available computing infrastructure.”

The following references are added:

⁶⁹Junior, P. E. F. et al. First-principles insights into the spin-valley physics of strained transition metal dichalcogenides monolayers. *New J. Phys.* 24, 083004 (2022).

⁷⁰Blundo, E. et al. Strain-Induced Exciton Hybridization in WS₂ Monolayers Unveiled by Zeeman-Splitting Measurements. *Phys. Rev. Lett.* 129, 067402 (2022).

⁷¹Raiber, S. et al. Ultrafast pseudospin quantum beats in multilayer WSe₂ and MoSe₂. *Nat. Commun.* 13, 4997 (2022).

⁷²Zollner, K. & Fabian, J. Engineering Proximity Exchange by Twisting: Reversal of Ferromagnetic and Emergence of Antiferromagnetic Dirac Bands in Graphene/Cr₂Ge₂Te₆. *Phys. Rev. Lett.* 128 (2022).

⁷³Zollner, K., Faria Junior, P. E. & Fabian, J. Strong manipulation of the valley splitting upon twisting and gating in MoSe₂/CrI₃ and WSe₂/CrI₃ van der Waals heterostructures. *Phys. Rev. B* 107, 035112 (2023)

We have updated the affiliations of the authors, and the Acknowledgements section, without affecting the scientific parts of the manuscript. We have also corrected some minor typos.